# Task-Specific Adaptation with Restricted Model Access

## Abstract

Modern foundation models achieve state-of-the-art performance across diverse modalities, yet commonly require modification of internal weights or insertion of new layers during fine-tuning. Such modifications increase deployment complexity, hinder optimization for edge devices, and risk exposure of proprietary model parameters. In this paper we analyze for the first time existing fine-tuning paradigms in the context of these three axes. Within this context we introduce "Gray-box" fine-tuning: a lightweight and deployment-friendly framework that adapts frozen backbones without altering their architecture or internal parameters. Gray-box fine-tuning enables adaptation solely via compact, external input/output adapters trained with controlled gradient signals at predefined model entry points, preserving all internal components unchanged. We introduce two variants: DarkGray-Box Adaptation (DGA), restricting modifications strictly to input and output interfaces, and LightGray-Box Adaptation (LGA), allowing limited injection of learnable tokens at intermediate layers for enhanced adaptability. Extensive evaluations across tasks including text-to-image retrieval, video retrieval, image classification, sketch retrieval, and diffusion-based generation demonstrate that Gray-box methods achieve competitive performance relative to standard fine-tuning, despite significantly stricter constraints. By decoupling task-specific adaptation from internal model modifications, Gray-box fine-tuning provides an efficient, scalable, and secure alternative to conventional fine-tuning methods.

## 1 Introduction

Recent advances in foundation models (Radford et al., 2021; Li et al., 2022; 2023; Oquab et al., 2023; Kirillov et al., 2023) have led to marked improvements in a variety of downstream applications. These models typically serve as pre-trained backbones, adapted to specific domains or tasks via fine-tuning. Although effective, current fine-tuning methods, including full fine-tuning (Devlin et al., 2019; Dosovitskiy et al., 2021), partial tuning (Girshick et al., 2014; Dosovitskiy et al., 2021), and parameter-efficient fine-tuning (PEFT) (Rebuffi et al., 2017; Hu et al., 2022; Lian et al., 2022; Zaken et al., 2022; Houlsby et al., 2019), still require injecting new layers inside the backbone model or retraining parameters. Such modifications introduce several practical shortcomings:

- **Scalability**: Maintaining separate adapted models for each specific task increases the complexity of deployment and resource use (Pope et al., 2023; Lester et al., 2021; Sheng et al., 2023; Gabrielsson et al., 2024).

- **Edge deployment**: Each adapted model may require separate optimization procedures for efficient edge deployment (*e.g.*, model pruning) (Lazarevich et al., 2021; Kwon et al., 2022).

- **Security/IP protection**: Modifying internal architectures or weights conflicts with model providers' need to safeguard proprietary information, highlighting the need for methods that enable adaptation while preserving model privacy (OpenAI, 2023; Haim et al., 2022).

To address these limitations, we introduce a new adaptation paradigm: "Gray-box" fine-tuning, a lightweight, scalable approach for effectively adapting foundation models without altering their internal structure or weights. Unlike black-box methods, which rely solely on inputs and outputs and thus offer limited adaptability, our gray-box techniques permit restricted access through gradient

propagation at carefully chosen entry points, such as model inputs or intermediate representations. Adaptation is achieved using lightweight, entirely *external* modules, preserving the integrity of the backbone model while ensuring safe, scalable, and efficient reuse.

We analyze two variants of gray-box adaptation. The first, **DarkGray-Box Adaptation (DGA)**, limits modifications to lightweight adapters at the input and output, with gradient access restricted exclusively to these endpoints. The second variant, **LightGray-Box Adaptation (LGA)**, extends this by additionally allowing the injection of learnable data tokens at specific intermediate layers. These variants, illustrated in Figure 1, address the earlier-discussed challenges of scalability, efficient edge deployment, and security by leveraging pre-trained foundation models without revealing or modifying their internal structure or weights.

Our novelty lies not in new adapter mechanics, but in our strict preservation of the model's *computational flow*: the fixed sequence of layer operations executed during inference. Many PEFT methods, though lightweight, alter this flow by inserting additional internal modules (*e.g.*, adapter layers) (Hu et al., 2022; Houlsby et al., 2019; Zaken et al., 2022). These alterations complicate deployment and increase infrastructure requirements. Recent methods, such as S-LoRA (Sheng et al., 2023) and Compress-then-Serve (Gabrielsson et al., 2024), attempt to mitigate these challenges specifically for LoRA adapters. However, they still necessitate additional mechanisms for managing multiple adapter modules within the backbone model. Our Gray-box approach avoids these complexities altogether by ensuring no modules are embedded within the backbone. The model remains sealed, preserving both efficiency and security. Deployment differences are illustrated in Figure 5, and Table 1 summarizes the benefits and trade-offs of various fine-tuning strategies.

Table 1: Comparison of different *"shades"* of fine-tuning methods. Each approach conceals different pieces of information regarding the backbone model and has varying requirements. The ✓ symbol indicates partial requirements that may vary depending on usage and often involves trade-offs.

| Approach | Hidden Information | | Deployment Requirements | | | | | | |
|---|---|---|---|---|---|---|---|---|---|
| | Gradients | Weights | Free of Layer Choice | Single Backbone Copy | Original Architecture | Original Weights | No Extra Layers | No Adapter Routing | Total #✓ |
| ☐ Full Fine-tune | ✗ | ✗ | ✗ | ✗ | ✓ | ✗ | ✓ | ✓ | 3.0 |
| ☐ LoRA | ✗ | ✓ | ✗ | ✓ | ✗ | ✓ | ✗ | ✗ | 2.0 |
| ☐ LGA (ours) | ✗ | ✓ | ✗ | ✓ | ✓ | ✓ | ✓ | ✓ | 6.0 |
| ☐ DGA (ours) | ✗ | ✓ | ✓ | ✓ | ✓ | ✓ | ✓ | ✓ | 7.0 |
| ■ Original (zero-shot) | ✓ | ✓ | ✓ | ✓ | ✓ | ✓ | ✓ | ✓ | 8.0 |

Our Gray-box approach has immediate practical implications. For instance, it significantly enhances deployment efficiency by allowing to maintain a single, shared backbone model and process large batches collectively, rather than distributing smaller batches across multiple separately adapted models. This unified approach substantially reduces complexity and resource usage (Pope et al., 2023; Sheng et al., 2023; Gabrielsson et al., 2024). Additionally, hospitals employing medical imaging models can securely enable third-party adaptations for specialized diagnostics without exposing sensitive model details (Bharati et al., 2022). While Federated Learning (FL)(Liu et al., 2024a) also addresses privacy concerns, it primarily protects data by distributing training across multiple nodes, typically requiring explicit knowledge of the model architecture for synchronization. In contrast, our Gray-box methods allow secure task adaptation while fully concealing both the architecture and weights. Additionally, foundation-model providers aiming to enable third-party adaptations can utilize our framework to gain many adaptations using a single, intact backbone, increasing efficiency while protecting their intellectual property.

**Gray vs. Black:** A common black-box adaptation approach involves training additional layers on top of a backbone's output features (Radford et al., 2021; Devlin et al., 2019; He et al., 2022; Oquab et al., 2023). However, such methods inherently underutilize the full expressive capability of the model due to their restricted interface. Our Gray-box framework addresses this limitation by enabling gradient-based adaptation at input or intermediate feature points, significantly enhancing the effectiveness of fine-tuning within limited-access constraints.

We evaluate our methods against four representative fine-tuning paradigms: (1) Full fine-tuning, (2) Last-layer fine-tuning, (3) LoRA (Hu et al., 2022) as a strong, common PEFT baseline, and (4) Black-box Linear Probing. The first two methods, which require full or partial access to the original weights, constitute white-box approaches. Our methods are tested across diverse tasks and backbone architectures, with LoRA and full fine-tuning serving as upper-bound references for achievable per-

Figure 1: An overview of our gray-box frameworks. **Left:** DarkGray-Box Input/Output Adapters (DGA) permits modifications only at the input and output levels while keeping the backbone model hidden and frozen. The only information available is the gradient flow (indicated by the orange-dotted arrow), which matches the shape of the last layer of the input adapter. **Right:** In contrast, LightGray-box (LGA) allows additional entry points into the model's intermediate layers, exposing slightly more information, such as the input dimensionality and the gradients of a subset of the layers.

formance. Our DarkGray-Box Input/Output Adapters (DGA) approach achieves particularly competitive results in retrieval tasks (*e.g.*, text-to-image and text-to-video retrieval benchmarks), as well as tasks less directly aligned with the model's original training domain, such as sketch-to-image retrieval and image classification. While we do not claim universal applicability across all possible models and tasks, our extensive evaluations demonstrate the robustness, flexibility, and practicality of our methods.

In summary, we offer the following contributions:

1. We introduce a new perspective for deployment-oriented adaptation paradigm, emphasizing the preservation of the original model's internal structure, weights, and computational flow, without compromising effective adaptation.
2. We propose two Gray-box methods balancing adaptation flexibility and model access:
   (a) **DarkGray-Box Adaptation (DGA)**: a minimal-access approach using lightweight adapters solely at input/output endpoints, maximizing deployment ease.
   (b) **LightGray-Box Adaptation (LGA)**: allows the additional injection of learnable tokens into intermediate layers, enhancing task performance while retaining model integrity.
3. We demonstrate competitive results of our methods across multiple modalities and domains, including retrieval, classification, and generation – comparing favorably against strong baselines like LoRA and full fine-tuning.
4. We conduct an extensive study to assess the individual and combined effectiveness of input and output adapters, providing detailed insights into their distinct roles and effectiveness.

## 2 RELATED WORK

**Prefix and Prompt Tuning** (Lester et al., 2021; Liu et al., 2021; Li & Liang, 2021) are methods proposed as lightweight alternatives to full fine-tuning for Large Language Models (LLMs). Instead of modifying all model parameters, these methods optimize a new set of input tokens for each NLP task. Prompt Tuning (Lester et al., 2021) focuses on optimizing a token sequence added to the first transformer's layer, while Prefix Tuning (Li & Liang, 2021) and Prompt Tuning 2 (Liu et al., 2021) propose optimizing a separate sequence added to each transformer layer. Due to unstable optimization when directly training prefix tokens, the Prefix-Tuning approach (Li & Liang, 2021) trains a matrix $P$, which is projected through a trainable MLP layer to compute the prefix added to the existing prompt input. Prefix-Tuning involves learning separate prefixes for both the encoder and decoder components of the LLM, inserted appropriately during inference. Depending on the task, these methods have proven effective with prefixes ranging from 10 to 200 learned tokens, along with their associated MLP layer. In this work, we simplify this approach by directly optimizing just two tokens for a single text encoder without additional components. Specifically, we use the first token as an attached prefix and the second as a "shift" token added to all original input tokens. Consequently, our approach increases the prompt's context length by only a single token per prompt or task, which

is particularly valuable for text encoders with limited context length (*e.g.* CLIP, which is limited to 77 tokens in total).

**Parameter-efficient fine-tuning (PEFT)** methods enable lightweight adaptation by freezing the backbone model and introducing small, trainable modules such as adapters Houlsby et al. (2019), BitFit Zaken et al. (2022), and LoRA Hu et al. (2022), among others (*e.g.*, Lian et al. (2022)). These modules typically modify the forward computation graph by injecting bottleneck MLPs, low-rank matrices, or bias-only updates, so that each downstream task attaches its own set of auxiliary layers. While this significantly reduces the number of trainable parameters, it introduces new deployment burdens: engineers must manage one shared backbone plus $N$ adapter modules and a runtime mechanism to load, merge, or swap them on demand. A prominent example is LoRA, which learns two $n \times r$ matrices whose product yields a low-rank update ($n \times n$) to the model's original weight matrices. These updates are then added to the frozen weights during inference. Recent works such as S-LoRA Sheng et al. (2023) and Compress-then-Serve Gabrielsson et al. (2024) aim to reduce this complexity by improving infrastructure for LoRA sharing and compression. Nonetheless, they still require dynamic orchestration across model variants and adapter bundles. While LoRA may appear "gray-box" due to weight freezing, recent work Horwitz et al. (2024) has shown that original model weights can be reconstructed from LoRA adapters, reclassifying it more accurately as a "white-box" method.

**Co-CoOp and MaPLe** A different lightweight fine-tuning approach is Co-CoOp (Zhou et al., 2022), a CLIP-based architecture designed to enhance the integration of visual and textual modalities for image classification. Co-CoOp concatenates the visual encoder outputs to the textual encoder input, conditioning the text on the image. Although Co-CoOp keeps CLIP frozen, this design requires both modalities during each inference, limiting the generation of non-conditioned textual feature vectors, an essential capability for tasks like Image Retrieval where query (text) and images (gallery) are encoded separately. Instead of input conditioning, MaPLe (Khattak et al., 2023) conducts prompt learning, inserted across different early stages layers of the CLIP textual and visual encoders, using learnable MLP network. MaPLe can be seen as an extension of Prefix-Tuning (Li & Liang, 2021) for classification tasks, freezing the model and allowing internal tokens to be learned, which respects the "LightGray-box" framework. We adapt a different version of this approach to our new tasks, where independent tokens are learned for each layer, with no shared or extra layers learned.

**Model thievery** has been widely studied in the context of neural networks (Tramèr et al., 2016; Krishna et al., 2020), with techniques ranging from replicating transformer behavior via output features (Sha et al., 2023) to reconstructing weights using gradients or known architectures (Milli et al., 2019; Horwitz et al., 2024; Béguelin et al., 2021). These attacks raise serious concerns about misuse and data leakage, as shown by efforts to extract training data from model weights (Haim et al., 2022; Bommasani et al., 2021). Our framework aims to reduce such risks by operating without access to internal weights or layers. While full recovery from gradient signals remains impractical, assessing the security bounds of Gray-box variants is left for future work.

In summary, "White-box" and "LightGray-box" methods have been explored in NLP and classification tasks by incorporating additional components or tokens into the model's intermediate layers. While input adapters have been studied in the context of LLMs, their application in the image domain has not been thoroughly investigated. We conduct this exploration through our LGA approach, which draws inspiration from these methods, and further develop a more restrictive DGA approach that preserves the original pretrained model's computational flow.

## 3 METHOD

In this section, we introduce our approach for fine-tuning a pretrained model $F$ (*e.g.*, foundation models CLIP, BLIP) for new domain-specific tasks without exposing its architecture or modifying its weights. We propose two fine-tuning settings, termed *DarkGray-box* and *LightGray-box* settings, both of which offer lightweight fine-tuning options, and leverage the pre-trained backbone model $F$ while handling the White-box challenges.

### 3.1 GRAY-BOX SETTINGS

**DarkGray-box:** In this setting, the internals of $F$ are completely hidden, akin to a black-box approach. The only exposed components are the *input* and *output adapters*, which are external train-

able modules plugged into the input and output of the backbone model. Formally, given an input $x$, the adapted model $\hat{F}$ output is defined as:

$$\hat{F}(x) := B \circ F \circ A(x) \tag{1}$$

where $A$ and $B$ are learned linear transformations applied exclusively at the model's input and output, respectively. To train these adapters, we require gradient propagation through the backbone $F$. Specifically, the gradient of the loss $\mathcal{L}$ with respect to the output of the input adapter $A(x)$ is accessible: $\nabla_A \mathcal{L} = \frac{\partial \mathcal{L}}{\partial \hat{F}} \circ B \circ \frac{\partial F}{\partial A}$. This means that a gradient tensor corresponding to the final layer of the input adapter is exposed — hence the term *DarkGray* instead of *Black*. Importantly, the backbone model's architecture and weights remain hidden, and only the adapters are trained. Our approach learns only a minimal number of parameters (approximately $0.4\%$ of the total model parameters). In this context, we address two types of input modalities: images and text.

**LightGray-box:** In this more relaxed setting, the provider introduces additional entry points where task-dependent information can be injected into the model's intermediate layers. This enables better adaptation to a domain-specific task, enhancing flexibility without compromising the advantages of the gray-box model setup. Specifically, we optimize a set of learnable tokens injected into the transformer layers of $F$, thereby influencing attention scores without accessing or modifying the weights or layers. Although this approach accesses the model's internal data paths, it preserves the internal architecture and weights hidden, retaining the advantages of a gray-box setting. It is important to note that while the model layers remain hidden, this setting requires access to their input tokens, and allowing gradients to propagate through them. Formally, we modify the standard attention mechanism: $\text{Attn}(K, Q, V) = \text{Softmax}\left(\frac{KQ^T}{\sqrt{d}}\right) V$ by introducing $k$ learnable proxy vectors $p_1, \ldots, p_k \in \mathbb{R}^d$ that are inserted into the key, query, and value spaces. The updated formulation becomes:

$$\text{Attn}(K, Q, V, p_1, \ldots, p_k) = \left[\text{Softmax}\left(\frac{K'Q'^T}{\sqrt{d}}\right) V'\right]_{1:m} \tag{2}$$

where $Q', K', V' \in \mathbb{R}^{(m+k) \times d}$ are created by vertically concatenating the original $Q, K, V \in \mathbb{R}^{m \times d}$ with the $k$ proxy tokens. The final output is sliced to retain only the original $m$ token positions, ensuring these proxy tokens influence only the current layer's attention scores and do not propagate into subsequent layers. Critically, this approach does not insert new layers; it simply provides additional learnable inputs to existing layers. During training, only the proxy tokens are optimized, leaving the underlying transformer layers and parameters entirely unchanged.

## 3.2 ADAPTERS

In this section, we outline a simple solution for the settings discussed above. Our DarkGray-Box Input/Output Adapters (DGA) setting transforms the original model's function $F(x)$ into $B \circ F \circ A(x)$, where $A$ and $B$ are *lightweight adapters* (linear operators), as opposed to modifying the function $F$ directly. We initialize $A$ and $B$ as the identity function to match $F(x) = B \circ F \circ A(x)$. The input adapter $A$ learns to transform the model's input into a representation that better aligns with task-specific requirements, while the output adapter $B$ applies a simple linear transformation to the model's output.

**Visual Input Adapter:** For image inputs, the visual adapter consists of learned 2D convolutional layers that preserve the original dimensions of the input image. Since no activation function is included, the visual adapter functions as an affine transformation on the image pixel space. As we observe later (in Section 5), this simplified visual adapter is sufficient for modifying the input for our purposes, and adding non-linear activations does not provide additional benefits.

**Textual Input Adapter:** For text inputs, we draw inspiration from previous works (Li & Liang, 2021; Lester et al., 2021; Liu et al., 2021) and train new textual tokens for the text encoder. However, unlike these methods, we find that optimizing just two tokens—the *extra* token and the *shift* token—is sufficient. The *extra* token is a learned token that is attached to the original input sequence. Due to the transformer's positional invariance (Vaswani et al., 2017), and the fact that positional encoding is not applied to this token, it can be flexibly inserted at any position within the input sequence. The *shift* token is another learned token that is added to each of the original input tokens, effectively "shifting" them within the token embedding space. Thus, this approach requires

only one extra token per prompt, which is particularly valuable for text encoders with limited context length (*e.g.* CLIP, which is limited to a total of 77 tokens).

**Output Adapters:** These adapters are applied to the model's output feature vector. For both image and text modalities, we implement the output adapters as simple linear layer on top of the feature vector space, similar to the linear probing approach (Oquab et al., 2023; Radford et al., 2021).

## 4 EVALUATION

We evaluate DGA and LGA across multiple tasks and benchmarks using various backbones, including CLIP-ViT-B/16, BLIP-B, and DINOv2-B. We compare their performance against the original model in the "Zero-Shot" (ZS) setting as a reference point (serving as a lower bound) and also against the Black-box Linear Probing (LP) baseline. Additionally, we compare them with three strong white-box alternatives that serve as upper bounds: Full Fine-Tuning (FT), Last Layers Fine-Tuning (LLFT), and LoRA, as discussed in Sections 1 and 2. Although FT involves the largest number of parameters, it often underperforms compared to lightweight approaches (*e.g.* LoRA, DGA) when the available training samples are insufficient for certain domains or tasks. Note that LLFT involves direct access to model layers, which places it in the white-box category. In Appendix A we conduct further evaluations on Text-To-Image diffusion, LLM and VLM backbones, for image generation, language understanding and image captioning, and also on CNN backbones. We report statistical significance tests comparing DGA and LGA in Appendix A.1 to validate performance differences. For full implementation details, please refer to Appendix E.

### 4.1 TEXT-TO-IMAGE RETRIEVAL

Table 2: Results on two Text-to-Image Retrieval datasets, using the BLIP backbone. The highest values are marked in **bold**, and the second best are underlined.

| Model | COCO 5k | | | | Flickr30K | | | |
|---|---|---|---|---|---|---|---|---|
| | R@1 | R@5 | R@10 | R@50 | R@1 | R@5 | R@10 | R@50 |
| Full FT | 53.06 | 79.32 | 87.58 | 97.62 | **87.3** | 96.5 | 98.1 | 99.4 |
| Last Layers FT | **54.32** | **80.32** | **87.66** | **97.68** | 86.5 | **96.7** | **98.3** | **99.7** |
| LoRA | 53.48 | 79.78 | 87.46 | 97.6 | 85.4 | 96.6 | 98.1 | 99.6 |
| **LGA (ours)** | 54.14 | 79.72 | 87.48 | 97.66 | 84.7 | 95.9 | 97.7 | 99.4 |
| MaPLe | 52.3 | 78.34 | 86.52 | 97.28 | 84.2 | 96.1 | 97.7 | 99.6 |
| **DGA (ours)** | 53.18 | 79.14 | 87.04 | 97.58 | 83.7 | 95.9 | 97.7 | 99.4 |
| Linear Probing | 51.4 | 78.28 | 86.26 | 97.52 | 83.5 | 95.6 | 97.6 | 99.3 |
| Original (ZS) | 47.04 | 74.18 | 83.1 | 96.36 | 78.5 | 94.5 | 96.8 | 98.9 |

Table 2 presents a comparison for fine-tuning BLIP on two Text-to-Image Retrieval benchmarks: COCO and Flickr30K. We observe that the LLFT baseline dominates in both datasets. LoRA, serving as a White-box upper bound, follows closely, while our Gray-box DGA shows a significant improvement with respect to zero-shot, and competitive performance to LoRA, with a recall@1 gap of only 0.30 points on COCO and 1.7 points on Flickr30K. Notably, DGA significantly improves over the ZS baseline, with a recall@1 increase of 6.14 points on COCO and 5.2 points on Flickr30K. LGA slightly improves DGA results by allowing multiple entries to the model's intermediate layers.

To further evaluate DGA and LGA on specific image domains, we created 12 distinct subsets of the COCO dataset using available human annotations to identify objects present in the images. Each subset includes all photos containing a specific element (*e.g.*, table, sky, sea) from both the training and test splits. Table 3 presents the results using the BLIP backbone. Notably, DGA consistently outperforms the ZS and LP baselines across all subsets, demonstrating the effectiveness of modifying the model's inputs and outputs. Additionally, the results demonstrate that LGA consistently outperforms DGA, emphasizing the advantages and flexibility of this more permissive configuration, which enables learning intermediate parameters/tokens. Interestingly, LoRA outperforms Full Fine-Tuning (FT) in most cases but is itself outperformed by the LLFT baseline, highlighting the influence of the number of optimized parameters relative to the dataset size. Our Gray-box approaches, DGA and LGA, together achieve top-2 performance in 58.33% (21/36) of cases, underscoring their competitive potential.

Table 3: Performance comparison using the BLIP backbone on different COCO sub-domain splits. Each domain corpus was collected based on annotated objects within the images (number of training images is in parentheses). Our adapters achieve performance on par with LoRA. The highest values are marked in **bold**, and the second best are underlined.

| | Building (23,021) | | | Furniture (17,882) | | | Grass (22,575) | | | Metal (22,526) | | |
|---|---|---|---|---|---|---|---|---|---|---|---|---|
| | R@1 | R@5 | R@10 | R@1 | R@5 | R@10 | R@1 | R@5 | R@10 | R@1 | R@5 | R@10 |
| Full Fine-tune | 58.47 | 84.54 | 91.18 | 62.51 | **88.59** | 93.48 | 65.2 | 88.76 | 94.97 | 61.8 | 85.2 | 91.53 |
| Last Layers FT | 60.06 | **85.73** | 91.77 | **63.09** | 87.54 | 93.58 | **68.42** | **91.43** | **95.82** | 62.08 | **86.72** | 91.6 |
| LoRA | 59.66 | 84.74 | **92.07** | 61.84 | 88.3 | 93.48 | 67.02 | 89.94 | 95.61 | **63.18** | 86.51 | **91.95** |
| **LGA (ours)** | **60.26** | 84.14 | 91.48 | 61.94 | 88.11 | **93.67** | 65.42 | 90.58 | 95.61 | 62.15 | 85.75 | 91.67 |
| MaPLe | 58.57 | 83.25 | 91.28 | 60.88 | 86.39 | 92.91 | 63.81 | 89.29 | 94.97 | 61.05 | 85.68 | 91.81 |
| **DGA (ours)** | 58.57 | 83.94 | 91.28 | 61.55 | 87.15 | 92.38 | 65.42 | 90.26 | 95.5 | 61.05 | 85.34 | 91.47 |
| Linear Probing | 56.89 | 83.35 | 90.98 | 60.98 | 86.1 | 92.14 | 64.67 | 89.72 | 94.97 | 59.26 | 84.65 | 90.64 |
| Original (zero-shot) | 52.63 | 80.77 | 87.41 | 56.76 | 83.99 | 90.7 | 59.53 | 87.47 | 93.79 | 56.23 | 81.83 | 89.26 |

| | Paper (9,521) | | | Pavement (18,311) | | | Road (15,402) | | | Sea (6,598) | | |
|---|---|---|---|---|---|---|---|---|---|---|---|---|
| | R@1 | R@5 | R@10 | R@1 | R@5 | R@10 | R@1 | R@5 | R@10 | R@1 | R@5 | R@10 |
| Full Fine-tune | 69.96 | 92.94 | **98.39** | 62.38 | 86.43 | 92.26 | 60.73 | 84.17 | 90.56 | 53.42 | 79.11 | 84.25 |
| Last Layers FT | 70.16 | **93.15** | 97.38 | **64.29** | 85.71 | 92.74 | 62.25 | 85.08 | 91.02 | 57.53 | 79.11 | 85.62 |
| LoRA | **71.57** | 92.94 | 96.98 | 63.33 | **87.14** | 92.74 | 60.73 | **85.54** | **91.32** | 54.45 | 80.14 | 84.59 |
| **LGA (ours)** | 70.97 | 92.54 | 97.18 | 62.74 | 86.9 | 92.26 | 61.19 | 84.78 | 91.02 | 56.51 | 80.14 | 83.9 |
| MaPLe | 70.16 | 92.54 | 96.37 | 62.38 | 86.19 | 92.38 | 59.97 | 84.02 | 89.95 | 55.48 | 79.11 | 83.9 |
| **DGA (ours)** | 70.56 | 91.73 | 96.57 | 61.55 | 86.9 | 92.14 | 61.04 | 83.56 | 90.56 | 55.82 | 78.42 | 84.59 |
| Linear Probing | 69.76 | 91.33 | 95.97 | 61.43 | 85.0 | 91.55 | 59.51 | 82.65 | 90.26 | 54.45 | 78.08 | 82.19 |
| Original (zero-shot) | 67.74 | 89.92 | 95.56 | 57.62 | 82.98 | 89.4 | 54.49 | 80.37 | 88.13 | 48.29 | 76.37 | 81.16 |

| | Sky (31,808) | | | Table (16,282) | | | Tree (36,466) | | | Window (14,209) | | |
|---|---|---|---|---|---|---|---|---|---|---|---|---|
| | R@1 | R@5 | R@10 | R@1 | R@5 | R@10 | R@1 | R@5 | R@10 | R@1 | R@5 | R@10 |
| Full Fine-tune | 57.06 | 83.91 | 91.46 | 65.3 | 89.18 | **94.99** | 57.05 | 83.74 | 90.23 | 70.91 | 92.87 | 96.53 |
| Last Layers FT | 59.42 | 84.82 | 91.69 | **66.36** | **90.5** | **94.99** | **59.02** | **85.11** | 91.15 | 71.87 | 93.45 | 96.92 |
| LoRA | 59.27 | **85.58** | 91.53 | 65.7 | 89.45 | 94.72 | 57.7 | 84.2 | **91.21** | **73.8** | **93.83** | 96.92 |
| **LGA (ours)** | **59.73** | 85.13 | 91.38 | 66.23 | 89.84 | 94.33 | 57.9 | 84.79 | 90.69 | 73.41 | 93.26 | **97.11** |
| MaPLe | 57.44 | 84.06 | 91.3 | 65.04 | 88.52 | 94.59 | 56.13 | 83.61 | 90.56 | 70.13 | 93.26 | 96.53 |
| **DGA (ours)** | 58.73 | 84.06 | 90.69 | 65.57 | 89.18 | **94.99** | 56.33 | 83.74 | 90.62 | 71.1 | 92.49 | **97.11** |
| Linear Probing | 56.98 | 83.6 | 90.39 | 62.4 | 87.6 | 94.06 | 55.54 | 83.15 | 90.1 | 68.79 | 92.49 | 96.53 |
| Original (zero-shot) | 52.78 | 80.32 | 87.72 | 59.37 | 83.25 | 92.74 | 51.15 | 79.67 | 88.26 | 67.44 | 91.33 | 95.57 |

Table 4: Precision@K comparison on the Stanford-Cars dataset, using the BLIP backbone. DGA is competitive with the strong white-box baselines, while outperformed by LGA across most metrics.

| | P@1 | P@5 | P@10 | P@50 | P@70 |
|---|---|---|---|---|---|
| Full FT | 98.07 | 98.08 | 97.76 | 77.64 | 57.55 |
| Last Layers FT | 95.03 | 95.8 | 95.99 | 76.02 | 57.13 |
| LoRa | 90.08 | 88.22 | 86.11 | 66.25 | 52.56 |
| **LGA (ours)** | **98.45** | **98.21** | 97.87 | **77.78** | 57.54 |
| MaPLe | 97.11 | 97.61 | 97.63 | 77.49 | 57.46 |
| **DGA (ours)** | 97.16 | 97.91 | **97.97** | 77.53 | **57.59** |
| Linear Probing | 78.1 | 74.9 | 74.38 | 55.73 | 45.96 |
| Original (ZS) | 63.96 | 62.67 | 58.51 | 40.73 | 34.78 |

Next, we conduct an experiment on the domain-specific Stanford-Cars dataset (Krause et al., 2013) as a retrieval task, which contains car images annotated by Make, Model, and Year (*e.g.*, "*2012 Tesla Model S or 2012 BMW M3 Coupe*"). Table 4 presents a Precision@K comparison using the BLIP backbone. Across all metrics, DGA and LGA significantly outperform both the ZS reference and the white-box baselines. Notably, the LoRA baseline underperforms compared to our methods, even though it still shows improvement over the ZS baseline. We attribute this phenomenon to the relatively low number of samples and specific vehicle descriptions (197) in the dataset, making adaptation in the input space more efficient. This suggests that the input adapter's flexibility offers an advantage in such cases. However, this trend is not consistent across all scenarios, as it may vary depending on the backbone model and the dataset used for training.

## 4.2 TEXT-TO-VIDEO RETRIEVAL

In Table 5 we test Text-to-Video Retrieval on two benchmarks: MSR-VTT and VATEX. We follow a previous approach Li et al. (2022) that applies Text-Image encoders at the frame level for video tasks. Following the established protocol, we uniformly sample 12 frames from each video and perform Text-to-Image Retrieval on the sampled frames. On both benchmarks, DGA achieves results

Table 5: Comparison on two Text-to-Video Retrieval benchmarks, using the BLIP backbone. Note that due to a small size of training set (7k videos), MSR-VTT full fine-tuning tends to underperform.

| Model | MSR-VTT | | | | VATEX | | | |
|---|---|---|---|---|---|---|---|---|
| | R@1 | R@5 | R@10 | R@50 | R@1 | R@5 | R@10 | R@50 |
| Full FT | 35.96 | 63.96 | 74.28 | 91.33 | **46.97** | **81.13** | **89.17** | **97.97** |
| Last Layers FT | 36.92 | 64.07 | 74.92 | 91.47 | 44.47 | 78.33 | 87.3 | 97.37 |
| LoRA | **37.72** | **65.77** | **76.27** | **92.31** | 41.63 | 75.43 | 84.43 | 96.5 |
| **LGA (ours)** | 37.04 | 64.14 | 74.29 | 91.36 | 41.23 | 75.43 | 83.6 | 95.67 |
| MaPLe | 35.17 | 61.33 | 71.9 | 89.79 | 39.03 | 71.53 | 82.27 | 95.37 |
| **DGA (ours)** | 37.24 | 63.98 | 74.21 | 91.34 | 41.03 | 73.2 | 82.8 | 95.53 |
| Linear Probing | 35.9 | 62.71 | 72.83 | 90.63 | 38.33 | 70.53 | 80.97 | 94.4 |
| Original (ZS) | 32.14 | 56.53 | 66.38 | 85.24 | 31.33 | 61.13 | 71.17 | 89.4 |

comparable to the LoRA baseline (*e.g.*, R@1 of 37.24% with DGA vs. 37.72% with LoRA), which performs best on MSR-VTT, with a recall@1 gap of less than one point and a difference of 1 to 2 points at higher recall@k levels. Moreover, DGA significantly outperforms the ZS reference, with a Recall@1 improvement of 5.1 points on MSR-VTT and 9.7 points on VATEX. It is notable that the white-box Full Fine-Tuning method outperforms all alternatives on VATEX but surpasses only the zero-shot and linear probing baselines on MSR-VTT. We attribute this to the combination of a high number of trainable parameters and the varying sizes of the training sets, with 26k videos in VATEX compared to only 7k in MSR-VTT. Evidently, the Last Layers Fine-Tuning baseline, with fewer trainable parameters, achieves better results than Full Fine-Tuning on the MSR-VTT dataset.

## 4.3 IMAGE CLASSIFICATION

Table 6: Image Classification results on two benchmarks, with CLIP. For ImageNet-1K, we trained with "16-shot" regime, where the training set was limited to 16 random images per class.

| Dataset | Accuracy | Original (ZS) | LP | **DGA (ours)** | MaPLe | **LGA (ours)** | LoRA | LL-FT | Full FT |
|---|---|---|---|---|---|---|---|---|---|
| ImageNet1k | Top-1 | 63.87 | 66.91 | 67.77 | 67.49 | 66.94 | 70.29 | 64.11 | **70.79** |
| | Top-5 | 87.82 | 90.23 | 91.66 | 90.83 | 90.45 | **92.81** | 87.31 | 92.29 |
| ImageNet Sketch | Top-1 | 46.97 | 57.30 | 60.06 | 54.57 | 67.48 | 69.04 | 80.97 | **81.05** |
| | Top-5 | 75.23 | 85.56 | 88.27 | 84.17 | 91.12 | 93.73 | **95.12** | 94.96 |

We further evaluate our approach on the Image Classification task using two benchmarks with a CLIP ViT-B/16 backbone, as shown in Table 6. The first classification task on ImageNet-1K (Russakovsky et al., 2014) while the second is sketch-domain classification on ImageNet-Sketch (Wang et al., 2019). For ImageNet-1K, we perform 16-shot training, sampling 16 images per class from the training set. DGA achieves a 3.9-point improvement in top-1 accuracy over the zero-shot baseline, while on the cross-domain ImageNet-Sketch, it gains a 13.1-point increase. However, LoRA outperforms DGA with a 2.52-point lead on ImageNet-1K and an 8.98-point lead on ImageNet-Sketch.

## 4.4 SKETCH-TO-IMAGE RETRIEVAL

Table 7: Sketch-to-Image Retrieval results on the Sketchy Dataset.

| | All Class | | | | Novel-Class-25 | | | |
|---|---|---|---|---|---|---|---|---|
| | R@1 | R@5 | R@10 | R@50 | R@1 | R@5 | R@10 | R@50 |
| Full FT | **69.20** | 91.44 | **96.08** | 98.80 | **34.92** | **60.44** | **71.80** | **90.56** |
| Last Layers FT | 65.52 | **91.60** | 95.92 | 98.80 | 30.44 | 56.92 | 68.60 | 90.12 |
| LoRA | 58.72 | 87.44 | 93.76 | **98.88** | 24.76 | 50.40 | 64.36 | 89.20 |
| **LGA (ours)** | 53.36 | 83.84 | 92.32 | 98.56 | 17.88 | 41.72 | 54.28 | 84.64 |
| **DGA (ours)** | 31.20 | 65.92 | 79.76 | 94.48 | 7.44 | 20.16 | 30.28 | 64.60 |
| Linear Probing | 21.12 | 57.92 | 73.84 | 92.24 | 3.72 | 12.80 | 19.44 | 49.08 |
| Original (ZS) | 8.56 | 26.64 | 40.16 | 64.08 | 1.80 | 6.76 | 10.64 | 34.80 |

Here we explore Instance Sketch-to-Image Retrieval experiment on the Sketchy dataset (Sangkloy et al., 2016). This dataset includes natural images paired with corresponding human-drawn sketches.

The goal is to retrieve *the exact original image* based on a given sketch (not just the class). For this task, we utilized the DinoV2 backbone, which has previously demonstrated strong image feature learning capabilities (Oquab et al., 2023). Notably, this backbone was trained on natural images, resulting in poor performance in the zero-shot setting, as shown in Table 7. Nonetheless, DGA achieves substantial improvement over the zero-shot baseline while keeping the backbone frozen and modifying only the input and output adapters. However, as this task involves adapting to a domain quite different from the original training domain, white-box methods like LoRA, Full FT, and LLF significantly outperform our approach due to their ability to modify model weights. Additionally, LGA, which can adjust internal attention scores, also outperforms DGA and LP by a large margin. These results together with ImageNet-Sketch suggest that in cross-domain settings, the model requires more substantial internal modifications, which limits the performance of the gray-box approach compared to white-box methods.

## 5  ABLATION STUDY

We analyze key components of DGA by measuring the contribution of each adapter using the CLIP model on the COCO 5k validation set, with zero-shot (ZS) performance as the reference. As shown in Table 8, adding either visual or textual input adapters ("DGA-I-vis", "DGA-I-txt") improves performance over ZS, and combining both input adapters ("DGA-I") yields a 5.72-point gain in R@1. Output adapters applied independently ("DGA-O-vis", "DGA-O-txt") also boost performance, and using both ("DGA-O") adds a further 5.74-point gain. We additionally test I/O adapters applied jointly to a single modality ("DGA-Text", "DGA-Vis"), which perform better than partial configurations. The best results are achieved with all I/O adapters enabled ("DGA"), confirming that modifying both input and output spaces across modalities is most effective. Further ablations, including analysis of learned prompt tokens (*e.g. shift* and *extra*), reveal that a single extra token is often sufficient, while adding more tokens may degrade performance due to reduced context length. We also study token placement and count in LGA. Full details and tables are provided in Appendix B.

Table 8: Ablation study on the COCO 5k validation set, with the CLIP model encoders.

| | Input Adapter | | Output Adapter | | Recall@K | | | |
|---|---|---|---|---|---|---|---|---|
| Baseline | Vision | Text | Vision | Text | R@1 | R@5 | R@10 | R@50 |
| Original (ZS) | ✗ | ✗ | ✗ | ✗ | 31.58 | 55.70 | 66.82 | 89.40 |
| DGA-I-txt | ✗ | ✔ | ✗ | ✗ | 35.78 | 62.02 | 72.90 | 92.70 |
| DGA-I-vis | ✔ | ✗ | ✗ | ✗ | 34.76 | 59.16 | 69.30 | 90.86 |
| DGA-I | ✔ | ✔ | ✗ | ✗ | 37.30 | 63.66 | 74.24 | 93.22 |
| DGA-O-txt | ✗ | ✗ | ✗ | ✔ | 40.76 | 67.72 | 78.18 | 95.18 |
| DGA-O-vis | ✗ | ✗ | ✔ | ✗ | 41.60 | 68.46 | 78.72 | 95.30 |
| DGA-O | ✗ | ✗ | ✔ | ✔ | 41.12 | 69.20 | 79.30 | 95.50 |
| DGA-Text | ✗ | ✔ | ✗ | ✔ | 40.92 | 68.62 | 79.00 | 95.32 |
| DGA-Vis | ✔ | ✗ | ✔ | ✗ | 41.88 | 68.74 | 78.72 | 95.10 |
| DGA | ✔ | ✔ | ✔ | ✔ | **43.04** | **70.52** | **80.26** | **95.94** |

## 6  SUMMARY AND LIMITATIONS

In this paper, we introduce Gray-box fine-tuning to address the deployment costs, security risks, and intellectual property challenges of conventional methods. Our two paradigms, **DarkGray-box Adaptation (DGA)** and **LightGray-box Adaptation (LGA)**, adapt frozen models using lightweight external modules. DGA restricts training to input/output adapters, fully preserving the model's internals, while LGA allows limited intermediate-layer access via learnable tokens for enhanced flexibility. Our methods achieve performance competitive with strong white-box baselines like LoRA across various tasks and modalities. However, our experiments also highlight that for cross-domain tasks with significant distributional shifts (*e.g.*, sketch-to-image), adaptation benefits from greater access to model internals. We believe our Gray-box methods represent an important step toward scalable adaptations for foundation model providers, using a single backbone.

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

# Appendix

This appendix provides additional details on our methods, experiments, and findings. We begin with further evaluations, including experiments on diffusion, LLM and CNN-based backbones, in Appendix A. In Appendix B, we conduct ablation studies on the number of input tokens in DGA and the choice of layers in LGA. Appendix C presents visualizations of the visual input adapter, offering insights into its transformations. Appendix D expands on recent developments in black-box prompt optimization and their limitations, along with a comparative analysis of task adaptation using input/output adapters. Finally, Appendix E details our experimental setup, including training configurations, hyperparameters, and model specifications.

## A  FURTHER EVALUATION

In this section, we conduct further evaluations on more tasks and backbones.

We extend our LGA approach to additional tasks across various backbones. We refrain from conducting full fine-tuning or last-layer fine-tuning due to resource constraints or pipeline incompatibilities (*e.g.* concatenation of multiple models).

Table 9: Evaluation of Text-To-Image Generation, using a pre-trained diffusion model.

|  | FID ↓ | LPIPS ↓ | CLIP-Similarity ↑ |
|---|---|---|---|
| Original (ZS) | 159.22 | 79.23 | 19.14 |
| LGA (Ours) | 87.78 | 77.99 | 20.84 |
| LoRA | 59.83 | 75.35 | 21.50 |

**Text-To-Image Generation:** We fine-tune a DiT-based diffusion model Chen et al. (2024) on the RSCID Lu et al. (2018) dataset, which consists of image-text pairs of satellite imagery—a domain previously shown to be underrepresented in web-scraped data Radford et al. (2021). Similar to other transformer-based tasks, we apply LGA entry points to the denoiser's attention layers. Table 9 presents results for both LoRA and our LGA approach, evaluating the generated images against the held-out test set using FID, LPIPS distance, and prompt adherence via the CLIP score. We observe a significant distribution shift between the fine-tuned models and the original, which was primarily trained to generate "natural" or "artistic" images. Figure 2 shows visual examples of generated images using multiple prompts, demonstrating that the fine-tuned models produce satellite imagery, which the original model is less likely to generate correctly.

**Image Captioning:** We fine-tune the BLIP-2 Li et al. (2023) backbone, using LGA, for the image captioning task. Table 10 presents results comparable to LoRA fine-tuning. The BLIP-2 backbone employs an image encoder followed by a Q-Former, which translates the prompt,including image tokens, into the token space of a frozen LLM. In this case, we were unable to optimize our DGA paradigm solely in the input space. The results indicate that our LGA achieves performance comparable to LoRA improving over the Zero-Shot.

Table 10: Image Captioning evaluation on the BLIP-2 backbone.

| Method | BLEU | BLEU Precision-1 | Length Ratio | Rouge1 | RougeLsum |
|---|---|---|---|---|---|
| Zero-Shot | 10.09 | 41.31 | 83.38 | 44.62 | 40.58 |
| LGA (ours) | **12.56** | 48.38 | **92.06** | 45.27 | 41.24 |
| LoRA | 12.41 | **48.91** | 90.23 | **45.36** | **41.39** |

**General Language Understanding Evaluation:** We fine-tune DeBERTa-v3-base LLM on the MRPC dataset, using LGA. Results are shown in Table 11 indicate again the LGA capability in finetuning to a new task even slightly outperforming LoRA.

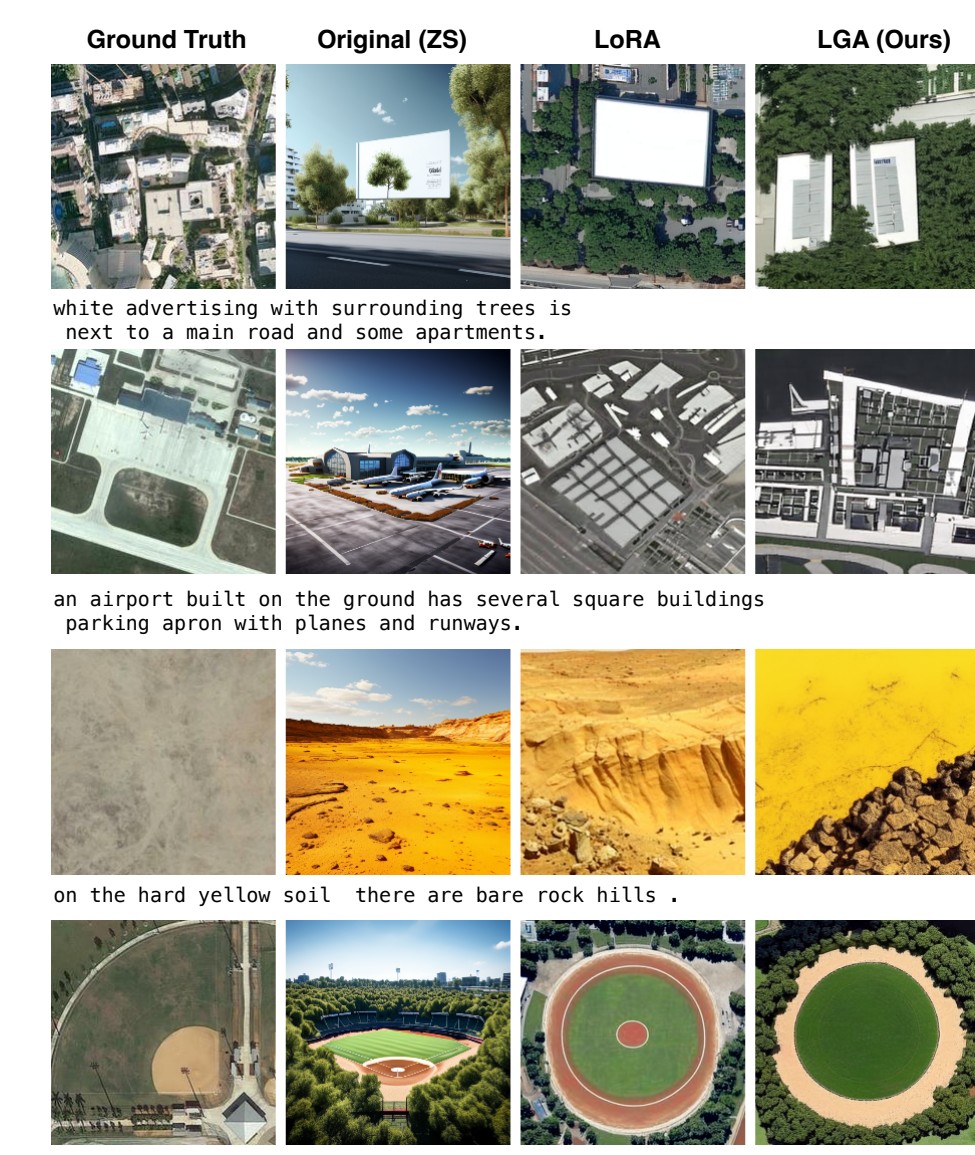

Figure 2: Generated images by three different model versions, of Original (zero-shot), LoRA and LGA.

Table 11: General Language Understanding Evaluation, on MRPC dataset with LLM Deberta-v3-base.

|          | Zero-Shot | LGA   | LoRA  | Full FT |
|----------|-----------|-------|-------|---------|
| Accuracy | 68.38     | 79.65 | 77.20 | 91.17   |

## A.1 STATISTICAL SIGNIFICANCE OF PERFORMANCE DIFFERENCES

To assess the robustness of observed performance trends between our proposed Gray-box methods DGA and LGA, we conduct statistical significance tests on key benchmarks. We ran five independent training and evaluation seeds for each method and computed p-values using paired t-tests for

Recall@K metrics. On COCO dataset in Table 2, LGA consistently and significantly outperforms DGA. The p-values for R@1, R@5, and R@10 are all below $1e-3$, confirming that the observed differences are statistically significant. On the Flickr30K benchmark, which is more saturated, differences between LGA and DGA are smaller. Still, the p-values for R@1, R@5, and R@10 are 0.020, 0.350, and 0.067, respectively, suggesting that LGA generally performs better, though DGA remains competitive. For Table 3: We further computed p-values across the 12 sub-domains. For R@1, 10 of the 12 categories showed statistically significant improvements ($p < 0.05$) for LGA over DGA. R@5 and R@10 trends are directionally similar but reflect more saturated recall regimes. A sample of p-values is provided in Table 12.

Table 12: P-values comparing LGA vs. DGA across COCO sub-domains (R@K), in Table 3.

|  | Building | Furniture | Grass | Metal | Paper | Pavement | Road | Sea | Sky | Table | Tree | Window |
|---|---|---|---|---|---|---|---|---|---|---|---|---|
| R@1 | 0.003 | 0.014 | 0.013 | 0.001 | 0.176 | 0.093 | 0.008 | 0.141 | 0.000 | 0.004 | 0.002 | 0.013 |
| R@5 | 0.130 | 0.002 | 0.010 | 0.208 | 0.041 | 0.463 | 0.010 | 0.036 | 0.003 | 0.014 | 0.024 | 0.023 |
| R@10 | 0.169 | 0.002 | 0.164 | 0.010 | 0.007 | 0.011 | 0.038 | 0.700 | 0.001 | 0.471 | 0.003 | 0.184 |

These results affirm that LGA provides statistically significant improvements over DGA in most settings. Nonetheless, DGA remains a competitive choice, especially in saturated tasks, and offers a compelling trade-off given its stricter Gray-box constraints.

## B    FURTHER ABLATION STUDY

In this section, we present additional ablation studies on the components of DGA and LGA. Table 13

Table 13: Ablation study on the number of optimized input tokens, in the text input adapter.

| Tokens # | R@1 | R@5 | R@10 | R@50 |
|---|---|---|---|---|
| **1** | 53.16 | 79.02 | 86.92 | 97.52 |
| **2** | 53.26 | 78.98 | 86.84 | 97.50 |
| **4** | 52.80 | 79.12 | 86.90 | 97.54 |
| **8** | 53.16 | 79.12 | 86.66 | 97.46 |
| **16** | 52.72 | 78.94 | 86.38 | 97.46 |
| **32** | 51.42 | 78.22 | 85.84 | 97.32 |
| **64** | 50.94 | 78.00 | 85.54 | 97.46 |
| **128** | 51.32 | 77.76 | 85.64 | 97.40 |

shows the ablation study on the number of input tokens optimized for the text encoder, with BLIP backbone. As observed, the optimal number of tokens lies between 1 and 8. However, it is not entirely clear which number is definitively optimal, as some metrics improve at the expense of others. For example, optimizing 2 tokens yields higher Recall@1 results compared to optimizing 1 token, but results in a lower Recall@5. Nevertheless, the differences across all token numbers are minimal, making their performance nearly on par. Consequently, we choose to optimize only 1 token to preserve the text-encoder context length from being occupied by these "proxy" tokens.

**CNN backbone:** Here we evaluate DGA on the following CLIP CNN-based models: CLIP-RN101, CLIP-RN50, CLIP-RN50x4, and CLIP-RN50x16. Table 14 presents the results on the COCO 5k validation set. Our DarkGray-box approach consistently improves upon the zero-shot (ZS) baseline across all backbones, although it remains inferior to the White-box Full Fine-Tuning (FT) baseline. We evaluate only these three approaches since these backbones are based on CNN architectures. While it is theoretically possible to apply LoRA to these CNN-based models, it is not straightforward due to the need to carefully select layers and adapt LoRA's implementation to CNN layers. Additionally, LGA is specifically tailored to transformer encoder architectures, making it unsuitable for these CNN backbones.

Table 15 presents a further evaluation of the CLIP backbone on the COCO subsets described in Section 4. We observe similar trends as with the BLIP backbone, where DGA consistently outperforms the Zero-Shot (ZS) and Linear Probing (LP) baselines. However, white-box methods that have access to model weights continue to outperform DGA and LGA, which leverage a frozen model.

Table 14: Evaluating DGA on all CLIP models based on CNN.

| Model # | R@1 | R@5 | R@10 | R@50 |
|---|---|---|---|---|
| CLIP-RN50 - FT | 43.64 | 72.34 | 82.22 | 96.12 |
| CLIP-RN50 - DGA | 32.92 | 60.50 | 72.36 | 92.76 |
| CLIP-RN50 - ZS | 26.46 | 50.30 | 61.58 | 86.88 |
| CLIP-RN101 - FT | 44.90 | 74.16 | 83.40 | 96.66 |
| CLIP-RN101 - DGA | 35.90 | 63.08 | 74.12 | 93.60 |
| CLIP-RN101 - ZS | 27.94 | 52.02 | 63.22 | 87.70 |
| CLIP-RN50X4 - FT | 47.28 | 76.42 | 84.72 | 97.02 |
| CLIP-RN50X4 - DGA | 38.74 | 66.40 | 76.64 | 95.04 |
| CLIP-RN50X4 - ZS | 31.12 | 54.62 | 65.70 | 89.30 |
| CLIP-RN50X16 - FT | 50.48 | 77.50 | 86.04 | 97.44 |
| CLIP-RN50X16 - DGA | 43.18 | 70.34 | 80.54 | 95.98 |
| CLIP-RN50X16 - ZS | 33.98 | 57.78 | 67.86 | 89.46 |

**Number of proxy tokens**: In Table 16, we conduct an ablation study on the choice of layers where the proxy vector is learned in LGA. This experiment is carried out on CLIP's visual encoder, trained on the COCO dataset. Injecting proxy vectors into the initial layers of the transformer encoder has a minimal effect, only slightly improving upon the zero-shot baseline, whereas the final layers have the most significant impact. However, using all transformer layers yields the best overall performance, eliminating the need for manual layer selection.

Next, examine the number of learned proxy vectors per layer in our LGA baseline, as presented in Table 17. Generally, increasing the number of learned vectors (and parameters) enhances the model's performance. However, we observe saturation in the Recall@10 and Recall@50 metrics starting from 8 learned vectors. It is important to note that as more vectors are learned, the gradient dimensionality required to propagate through the model to the learned parameters increases, resulting in a trade-off with the amount of information exposed in the Gray-box approach.

In Table 19 we ablate over the number of BLIP last layers fine-tuning. Each model was trained on COCO training set, results presented on COCO 5k validation set. We observe minor differences on performance between the methods, where fine-tuning all the layers results in lower performance. We relate it to the high number of parameters versus the low size of training set.

## C  VISUALIZATION

In this section, we visualize the image transformations produced by the input adapter. Figure 3 shows randomly sampled images from the COCO dataset. Each original image is processed through the input adapter and normalized to the same mean and standard deviation as the original image for visualization. Although the transformed images may appear corrupted or unnatural to the human eye, the model interprets these modified versions more effectively, as evidenced by performance improvements across multiple benchmarks.

## D  FURTHER DISCUSSION ON RECENT STUDIES

Recent studies Liu et al. (2024b); Wang et al. (2024) have proposed black-box prompt optimization techniques for Vision-Language models, aiming to enhance performance without requiring access to the backbone model. These methods achieve this by optimizing the input textual prompt, focusing exclusively on text manipulation Wang et al. (2024) or text-to-text mapping Liu et al. (2024b), without addressing the visual modality. More specifically, they are designed to optimize textual prompts for tasks such as 16-shot classification. However, this approach limits their applicability to scenarios heavily reliant on the visual domain. For instance, tasks such as Video or Sketch retrieval, which are fundamentally based on visual inputs, remain outside the capabilities of these methods. In contrast, our work addresses such visual domain challenges, expanding the utility and applicability of black-box fine-tuning to a broader range of tasks beyond text-focused optimizations.

Table 15: Performance comparison using the CLIP backbone on different COCO sub-domain splits. Each domain corpus was collected based on human-annotated objects within the images (number of training images in parentheses). Our adapters achieve performance on par with LoRA.

| | Building (23,021) | | | Furniture (17,882) | | | Grass (22,575) | | | Metal (22,526) | | |
|---|---|---|---|---|---|---|---|---|---|---|---|---|
| | R@1 | R@5 | R@10 | R@1 | R@5 | R@10 | R@1 | R@5 | R@10 | R@1 | R@5 | R@ |
| Full Fine-tune | 47.18 | 77.6 | 87.22 | 48.9 | 78.81 | 87.92 | 53.75 | 83.4 | 91.54 | 49.35 | 76.46 | 85.8 |
| Last Layers FT | **54.11** | 80.08 | **89.89** | 56.57 | **83.7** | 90.51 | 58.78 | **87.26** | **94.0** | **56.16** | **82.38** | 90.2 |
| LoRA | 53.32 | **80.77** | 88.4 | **58.2** | 83.51 | **91.28** | **59.21** | 86.08 | **94.0** | 55.61 | 80.87 | 89.0 |
| **LGA (ours)** | 52.43 | 78.79 | 87.41 | 56.95 | 82.36 | 90.12 | 55.46 | 84.9 | 92.72 | 54.3 | 79.49 | 87.6 |
| MaPLe | 49.36 | 76.81 | 85.93 | 55.7 | 80.54 | 89.07 | 53.75 | 83.3 | 92.29 | 53.34 | 78.53 | 86.5 |
| **DGA (ours)** | 49.75 | 75.62 | 83.85 | 52.73 | 79.29 | 88.69 | 52.03 | 81.26 | 89.72 | 49.55 | 76.19 | 84.3 |
| Linear Probing | 46.78 | 73.24 | 83.55 | 52.83 | 78.91 | 87.34 | 52.03 | 80.19 | 89.83 | 48.86 | 75.43 | 84.7 |
| Original (zero-shot) | 35.88 | 61.15 | 71.75 | 44.68 | 70.66 | 79.77 | 40.36 | 67.67 | 80.62 | 40.67 | 65.79 | 75.6 |

| | Paper (9,521) | | | Pavement (18,311) | | | Road (15,402) | | | Sea (6,598) | | |
|---|---|---|---|---|---|---|---|---|---|---|---|---|
| | R@1 | R@5 | R@10 | R@1 | R@5 | R@10 | R@1 | R@5 | R@10 | R@1 | R@5 | R@ |
| Full Fine-tune | 59.68 | 85.69 | 93.35 | 54.76 | 81.43 | 87.86 | 48.86 | 80.21 | 87.37 | 43.15 | 69.86 | 80.8 |
| Last Layers FT | **65.12** | **89.11** | **95.77** | 57.98 | 83.1 | 88.93 | 56.77 | 79.91 | **89.19** | 48.97 | **75.34** | 82.1 |
| LoRA | 63.51 | 88.1 | 94.76 | **61.55** | **84.52** | **90.24** | **58.75** | 81.58 | **89.19** | **49.66** | 75.0 | 81.5 |
| **LGA (ours)** | 61.29 | 87.7 | 94.35 | 59.52 | 82.5 | 89.05 | 55.86 | 80.82 | 88.13 | 47.95 | 74.32 | 80.8 |
| MaPLe | 59.27 | 87.9 | 93.75 | 57.98 | 80.0 | 88.69 | 54.34 | 79.0 | 87.52 | 46.92 | 71.58 | 78.4 |
| **DGA (ours)** | 59.07 | 86.29 | 92.94 | 53.33 | 79.4 | 85.71 | 53.58 | 77.17 | 85.39 | 40.75 | 70.55 | 80.8 |
| Linear Probing | 58.87 | 85.48 | 91.94 | 52.38 | 78.93 | 86.07 | 50.84 | 77.02 | 83.71 | 42.81 | 70.89 | 78.4 |
| Original (zero-shot) | 52.62 | 77.42 | 86.69 | 40.95 | 65.95 | 76.31 | 38.51 | 62.71 | 73.36 | 36.3 | 59.93 | 71.2 |

| | Sky (31,808) | | | Table (16,282) | | | Tree (36,466) | | | Window (14,209) | | |
|---|---|---|---|---|---|---|---|---|---|---|---|---|
| | R@1 | R@5 | R@10 | R@1 | R@5 | R@10 | R@1 | R@5 | R@10 | R@1 | R@5 | R@ |
| Full Fine-tune | 47.44 | 77.12 | 87.87 | 55.15 | 81.53 | 88.65 | 46.43 | 77.9 | 86.36 | 56.07 | 85.16 | 91.3 |
| Last Layers FT | **52.1** | **81.92** | **90.31** | 58.71 | **85.62** | 91.82 | 53.05 | **81.9** | **89.38** | **65.9** | 88.44 | 94.6 |
| LoRA | 51.33 | 80.32 | 89.24 | **59.63** | 83.25 | **92.35** | **53.11** | 80.52 | 88.79 | 64.93 | **88.63** | 95.3 |
| **LGA (ours)** | 49.43 | 78.49 | 87.87 | 57.39 | 83.77 | 91.42 | 50.49 | 79.87 | 87.08 | 64.35 | 88.05 | 95.1 |
| MaPLe | 48.51 | 77.04 | 87.57 | 58.18 | 82.98 | 89.84 | 47.61 | 77.38 | 86.03 | 63.2 | 87.48 | 94.0 |
| **DGA (ours)** | 45.16 | 75.9 | 85.43 | 54.22 | 81.13 | 88.52 | 47.15 | 75.8 | 84.66 | 59.92 | 86.51 | 93.0 |
| Linear Probing | 43.17 | 73.91 | 84.82 | 53.56 | 80.87 | 88.65 | 45.31 | 74.56 | 83.34 | 59.92 | 86.13 | 93.0 |
| Original (zero-shot) | 34.86 | 61.4 | 73.07 | 44.46 | 72.3 | 80.47 | 35.74 | 61.64 | 73.31 | 50.87 | 80.15 | 87.8 |

Table 16: Ablation study on choice of layers in for the proxy vectors.

| Layers # | R@1 | R@5 | R@10 | R@50 |
|---|---|---|---|---|
| **No FT (zero-shot)** | 42.02 | 69.28 | 79.34 | 95.02 |
| **First layers (0-3)** | 43.10 | 70.16 | 80.08 | 95.80 |
| **Middle layers (4-7)** | 44.56 | 71.22 | 81.20 | 96.16 |
| **Final layers (8-11)** | 44.76 | 71.80 | 81.58 | 96.36 |
| **All layers (0-11)** | 44.88 | 72.56 | 81.98 | 96.26 |

To further illustrate the broader applicability of our approach, Figures 4 and 5 present a demonstration of general schemes for handling multiple tasks or domains. The bottom part of the figure illustrates the naive approach of managing each task or domain with its own optimized model. In contrast, the top part of the figure shows a single optimized backbone model capable of handling all inputs with the use of input/output adapters. First, each input is processed using the appropriate lightweight input adapter. Next, the aggregated batch across all tasks is fed into the model, which produces outputs for each item. Finally, each output is post-processed with its corresponding output adapter to generate the final result.

Table 17: Ablation study on the number of learned proxy vector per layer in LGA, on the CLIP backbone.

| Tokens # | R@1 | R@5 | R@10 | R@50 |
|----------|-------|-------|-------|-------|
| 1 | 44.54 | 71.80 | 81.42 | 96.16 |
| 2 | 44.60 | 72.28 | 81.88 | 96.12 |
| 4 | 45.40 | 72.12 | 81.98 | 96.32 |
| 8 | 45.46 | 72.82 | 82.44 | 96.22 |
| 16 | 46.08 | 73.32 | 82.46 | 96.34 |
| 32 | 46.12 | 73.50 | 82.46 | 96.44 |
| 64 | 46.42 | 73.68 | 82.40 | 96.36 |

Table 18: Ablation study on the textual input adapter components, shift and extra token, on the CLIP backbone.

| Token | R@1 | R@5 | R@10 | R@50 |
|-------|-------|-------|-------|-------|
| **Only Extra** | 35.32 | 61.28 | 72.08 | 92.14 |
| **Only Shift** | 33.92 | 59.28 | 70.52 | 91.46 |
| **Both** | 35.80 | 61.34 | 72.30 | 92.54 |

**Experimental Validation**: To substantiate these claims, we conducted inference experiments comparing two setups: 1) A single backbone combined with 10 pairs of DGA adapters (for 10 different tasks or domains), 2) Ten separate backbones without using our DGA framework. In each setup, we utilize CLIP encoders to encode 10 sampled sets of 100 pairs of images (224x224) and their captions, a total of 1,000 paired samples.

The results demonstrate significant computational and memory efficiency with our approach: Our framework required 22.760 GFLOPs for 1000 samples, compared to 203.223 GFLOPs for the separate backbone setup. Similarly, GPU memory usage was reduced to 1.462 GB, as opposed to 14.54 GB in the alternative setup. These results highlight the resource efficiency and scalability of our framework in managing diverse tasks or domains.

# E    IMPLEMENTATION DETAILS

This section provides the implementation details of our experiments. Figure 6 provides an overview of our input adapters. All methods are trained using the *AdamW* optimizer, with training conducted on 1-4 nodes of *NVIDIA A100* GPUs, depending on the batch size. The input/output adapters are initialized as identity functions.

**Learning Rates**: For CLIP backbones, we train DGA with an initial learning rate of $1 \times 10^{-4}$, and $5 \times 10^{-5}$ for BLIP and DinoV2, all with an exponential decay rate of $0.93$ down to a minimum of $1 \times 10^{-6}$.

**Batch Sizes**: We use a batch size of 256 for all retrieval tasks, except for the Stanford-Cars dataset, where a batch size of 64 is applied. For ImageNet1k classification, a batch size of 1024 is used, and 64 for ImageNet-Sketch.

**Epochs:** We train the models for the following number of epochs on each benchmark: 25 for Stanford-Cars and ImageNet1k (16 shots), 30 for Sketchy and ImageNet-Sketch, 50 for COCO, 2 for Flickr30k, 20 for MSR-VTT, and 40 for VATEX.

**LoRA Hyper-parameters**: For the LoRA baseline, we adapt the $Q$, $K$, and $V$ matrices across all transformer layers, ensuring the rank matches the number of parameters used by DGA and LGA, depending on the backbone.

**Trainable Parameters**: The number of trainable parameters depends on the backbone. For BLIP-B, DGA optimizes 0.10% of the parameters, 0.42% for CLIP, and 1.57% for DINOv2. To ensure a fair comparison, we train the LoRA baselines with a rank $r$ that results in a matched number of trainable

Table 19: Ablation study on number of the BLIP last layers fine-tuning, on the COCO dataset.

| Layers # | R@1 | R@5 | R@10 | R@50 |
|---|---|---|---|---|
| **1** | 54.12 | 80.36 | 87.74 | 97.72 |
| **2** | 54.16 | 80.74 | 87.64 | 97.86 |
| **3** | 54.22 | 80.64 | 88.00 | 97.80 |
| **4** | 54.16 | 80.78 | 87.88 | 97.74 |
| **5** | 53.60 | 80.30 | 88.02 | 97.74 |
| **All** | 53.86 | 79.62 | 87.88 | 97.62 |

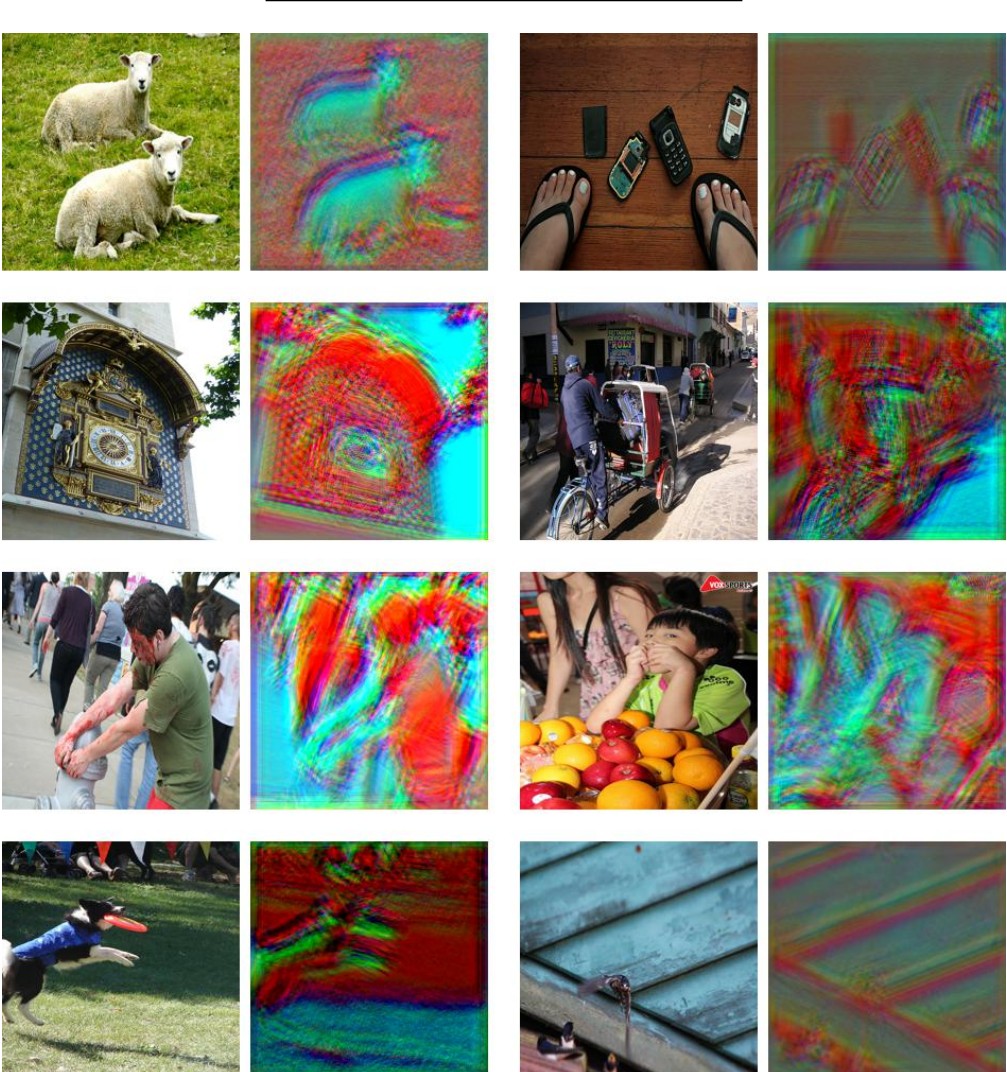

Figure 3: Visualization of the input adapter's influence on images.

parameters to DGA: $r = 8$ for CLIP, $r = 2$ for BLIP, and $r = 25$ for DINOv2. For LGA, we train a proxy token for each of the 12 transformer layers, resulting in a maximum of $12 \cdot 2 \cdot 768$ trainable parameters, depending on the backbone's dimensionality and the number of modalities (image and text).

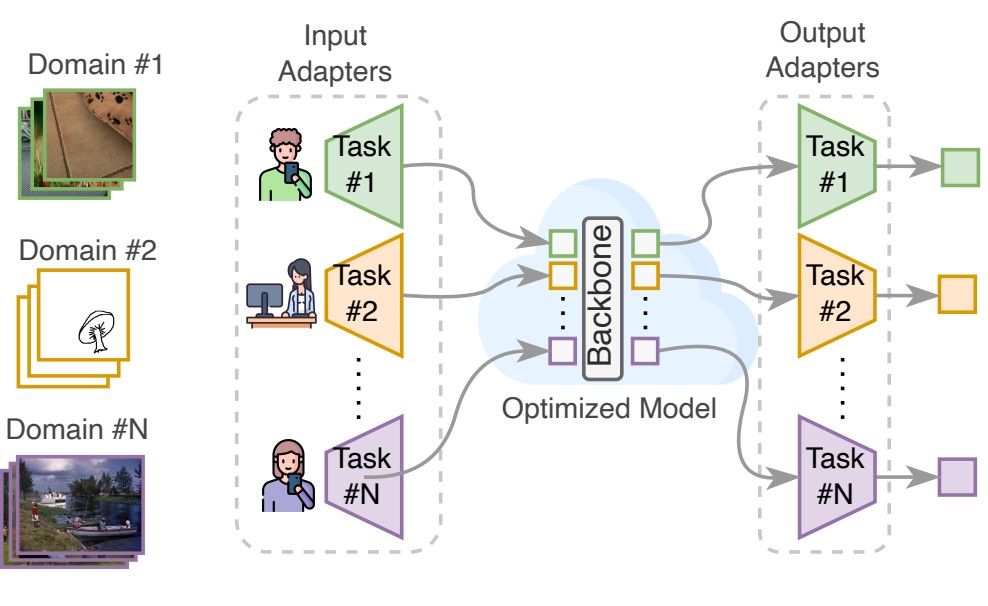

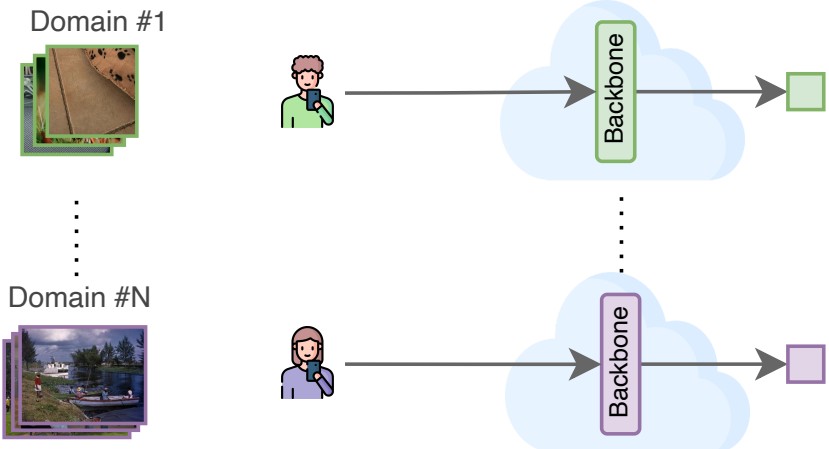

Figure 4: General schemes for handling $N$ different tasks or domains. **Top**: A single optimized model designed for multiple tasks or domains. **Bottom**: A naive approach with $N$ different models, one for each task.

**Our DGA: Computational Flow remain Unchanged**

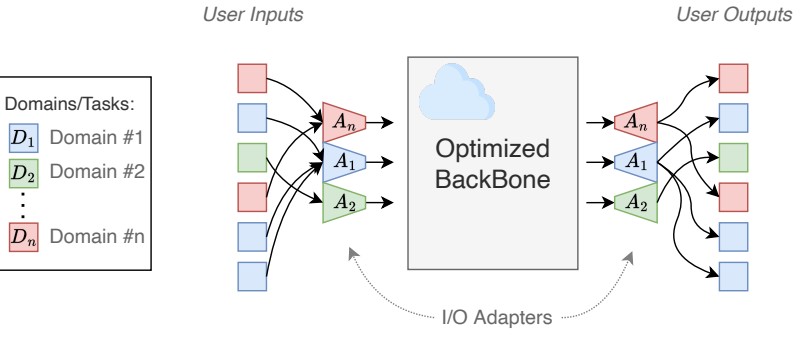

**PEFT Methods: changing computational flow**

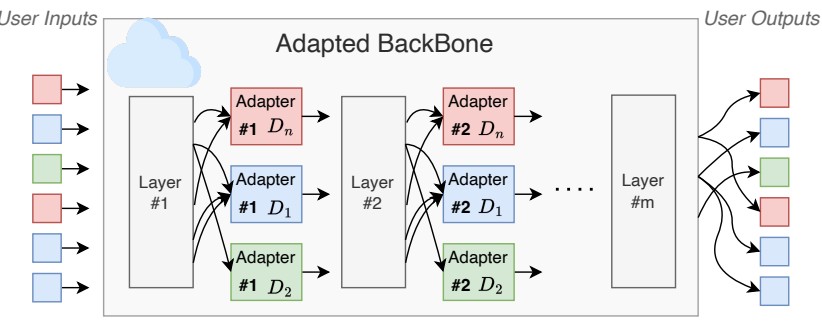

**Our LGA: Computational Flow remain Unchanged**

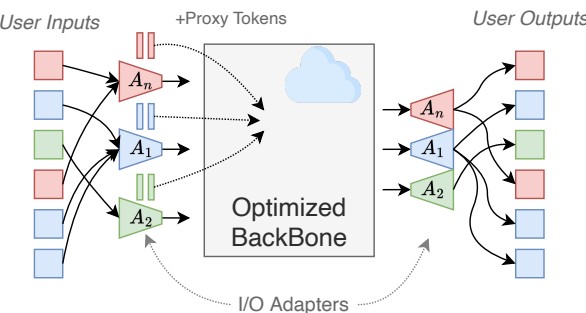

Figure 5: Deployment visualization schemes of DGA (Top), general PEFT methods (Middle) and LGA (Bottom). Internal adapters would alter the original computational flow, potentially breaking hardware-level optimizations and reducing efficiency (middle).

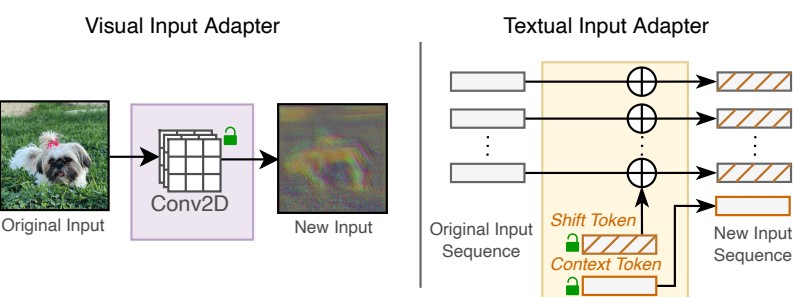

Figure 6: An overview of our Input Adapters. The visual input adapter (left) consists of 2D task-specific convolutional layers that preserve the image's original size. The textual input adapter (right) includes two task-specific tokens: a "shift" token added to the original sequence tokens and an "extra" token appended to the original sequence as a contextual token. Both adapters transform the original input into a new representation that better aligns with the pre-trained backbone model.

