# OpenReview forum: "Task-Specific Adaptation with Restricted Model Access"
_ICLR.cc/2026/Conference — Submitted to ICLR 2026_

### Official Review · Reviewer_Rr9e · 2025-10-27

**Soundness:** 2
**Presentation:** 2
**Contribution:** 2
**Rating:** 4
**Confidence:** 3

**Summary:**

The paper formalizes a gray-box fine-tuning regime for adapting frozen backbones under strict access constraints. Two variants are proposed: DGA (input/output adapters only) and LGA (a few learnable tokens injected at selected layers without editing backbone weights). The pitch is deployment/IP friendliness, where one sealed backbone is reused across tasks via small per-task attachments, while remaining competitive (though not universally SOTA) on text–image/video retrieval and classification.

**Strengths:**

(1) Deployment relevance: Clear, enforceable access model (sealed backbone; adapter/token attachments) aligns with multi-tenant serving and IP concerns.
(2) Crisp constraint definition: “No weight edits; preserve computational flow” makes the scope precise and implementable.
(3) Competitive under constraints: Meaningful gains over zero-shot and linear probes; often close to LoRA/last-layer FT despite stricter access limits.
(4) Backbone diversity: Results include both ViT and CNN (e.g., CLIP RN101), supporting generality.
(5) Ablations & design rationale: Decomposes input vs. output adapters and token count/placement; clarifies which components matter.
(6) Transparency about trade-offs: Acknowledges cases where LoRA/FT win.

**Weaknesses:**

(1) Missing closest SOTA baselines: Lacks comparison with CLIP-Adapter and Tip-Adapter and recent constrained CLIP prompt learners, with tuning parity and modern training recipes.
(2) Non-SOTA outcomes under-justified: When underperforming LoRA/FT, the paper lacks causal analysis (e.g., capacity limits of I/O adapters, architectural locality, data regime effects) and a quantitative cost–benefit (accuracy vs. trainable params, GPU-hours, latency, memory, concurrency).
(3) Reporting gaps: Lack of disclosure of trainable parameter counts, compute budgets, hyperparameter search spaces/budgets for all methods, and also a lack of broader distribution-shift tests.

Minor
(4) Editorial issues: Repetition of full forms (e.g., expanding “Dark-Gray-box Adaptation (DGA)” multiple times). Define it once, then use the acronym.

**Questions:**

Along with the weaknesses raised, here are a few questions that needs to be addressed

1. Closest baselines: Why are CLIP-Adapter, Tip-Adapter/Tip-Adapter-F, VPT (and recent constrained prompt learners) not included? Please add them with tuning parity and modern training recipes.
2. Trade-off analysis: For cases where you trail LoRA/FT, provide a causal diagnosis (capacity limits, layer-wise adaptation needs, data regime) and a quantitative trade-off table (acc. vs. trainable params/GPU-hours/latency/memory/concurrency).

---

> ### Author Response · Authors · 2025-11-27
>
> We thank the reviewer for highlighting the strengths of our work, including the clarity of the constraint model, the deployment/IP relevance, the backbone diversity, and the thorough design ablations. Below we address the raised concerns, also answer the reviewer’s questions:
>
> ## 1. Missing Closest SOTA Baselines
> Our baselines were selected to emphasize comparisons **across different fine-tuning paradigms and gradient-access constraints**, as explicitly described in lines 104–129. We chose LoRA because it is the most widely adopted PEFT baseline operating under similar gradient requirements. Importantly, we do not claim SOTA performance—our goals differ, as you noted in your strengths, and our focus is the access model, not chasing maximal accuracy.
> Regarding the specific baselines mentioned (CLIP-Adapter, Tip-Adapter, VPT, etc.), as we discuss in lines 164–179, these methods **alter the backbone’s computational graph by inserting additional components**, which directly violates the gray-box constraint of keeping the backbone sealed.
> ## 2. Non-SOTA Outcomes and Causal Interpretation
> We appreciate the reviewer raising this point. However, we would like to clarify that the causal factors behind GrayBox’s performance relative to LoRA/FT are already analyzed explicitly in the main paper (Section 4.4, lines 432-443; Section 6, lines 483-485). In these sections, we discuss that gray-box methods intentionally operate under strict capacity and access constraints, some downstream tasks require deeper layer-wise adaptation that our model intentionally avoids, and large distribution shifts inherently favor methods that modify internal weights.
> Regarding the request for a cost-benefit analysis, we note that our framework does not claim training-time or compute advantages over LoRA/FT. The motivation for gray-box adaptation is fundamentally different: it is an access-regime design aimed at sealed-backbone deployment scenarios, rather than a method for reducing GPU-hours or increasing accuracy under white-box conditions. That said, to further support our scalability claim, we have a GFLOPs comparison in Appendix D, showing our adapters are substantially efficient for many adaptation tasks.
> ## 3. Reporting Gaps (parameter counts, compute budget, hyperparameter search, distribution shift tests)
> The following information **is reported in Appendix E**, including trainable parameter counts, adapter sizes, hyper-paramters search (follows the same fixed grid for all PEFT baselines), etc. Regarding “a lack of broader distribution-shift tests” - In addition to the experiments in **Section 4.4**, we conducted further experiments (e.g., diffusion model natural images to satellite imagery) in **Appendix A**.
>
>
> ## 4. Editorial Issues
> We thank the reviewer for pointing this out. We will clean up acronym usage and remove redundant expansions.
>
> We thank reviewer Rr9e once again for the constructive feedback and for recognizing the deployment relevance, clarity of our constraint model, and strength of our experimental and ablation studies.

---

### Official Review · Reviewer_ka9w · 2025-10-30

**Soundness:** 3
**Presentation:** 2
**Contribution:** 3
**Rating:** 4
**Confidence:** 5

**Summary:**

This paper introduces a paradigm for adapting large foundation models without modifying their internal weights or architecture. The core idea is to maintain a frozen, sealed backbone model and perform adaptation solely through lightweight, external modules. The authors propose two specific instantiations: DarkGray-Box Adaptation (DGA), which restricts modifications to input and output adapters, and LightGray-Box Adaptation (LGA), which additionally allows the injection of learnable tokens into intermediate layers of a transformer to influence attention mechanisms.

**Strengths:**

- Strong Empirical Validation: The paper is thorough in its experiments. Evaluating across multiple tasks (retrieval, classification, generation), modalities (image, text, video, sketch), and model architectures (ViT, CNN) provides convincing evidence for the generality and robustness of the proposed methods. The consistent trend showing DGA/LGA's competitiveness with LoRA is impressive.

- Comprehensive Analysis: The inclusion of detailed ablation studies, the number of proxy tokens, layer selection for LGA) is commendable. It provides deep insights into what makes the method work and offers practical guidance for future implementations. The statistical significance tests add rigor to the claims.

**Weaknesses:**

1. Limited theoretical or empirical analysis: While the paper's motivation heavily leans on security and IP protection, the evaluation of these aspects is relatively superficial. The authors correctly note that LoRA weights can be used to recover original weights, thus re-classifying it as white-box. However, they do not provide a similar security analysis for their own methods. Could the gradients exposed in DGA, or the intermediate activations and gradients in LGA, be exploited in a model extraction or data reconstruction attack? A discussion, even if preliminary, on the security bounds of the "Gray-box" is a significant missing piece.

2. Weak performance in significant domain shifts: The results on the Sketchy dataset and ImageNet-Sketch clearly show that Gray-box methods struggle compared to white-box methods when the target domain is vastly different from the pre-training domain (e.g., natural images vs. sketches). This is an important limitation that should be more prominently discussed in the main text, as it delineates the boundaries of the method's applicability. It suggests that internal weight adjustments are still necessary for radical domain adaptation.

3. Positioning against very recent black-box methods: The discussion in Appendix D is a good start, but the paper could more forcefully position itself against the latest "prompt optimization" or "black-box tuning" literature (e.g., methods that optimize only the input text prompt). The claim that these methods are limited to text is valid, but a more direct comparison on a shared task (e.g., 16-shot classification) would strengthen the argument that Gray-box access provides a tangible advantage over pure black-box approaches.

4. Computation and efficiency analysis: The approach requires full gradient back-propagation through the frozen model for both DGA and LGA. While the number of trained parameters is low, the training time and memory footprint are likely closer to full fine-tuning than to inference-time prompt tuning methods. A brief discussion of training efficiency would provide a more complete picture.

**Questions:**

As mentioned above

---

> ### Author Response · Authors · 2025-11-27
>
> We thank reviewer ka9w for recognizing the breadth of our empirical evaluation, the depth of our ablations, and the statistical rigor of our analysis. We address the raised concerns below:
> ## Security / IP-Protection Motivation and Analysis
> We would like to clarify that **our paper does not lean heavily on security or IP protection as its primary contribution**. Instead, these considerations serve as one motivating scenario among several (others include deployment simplicity, single-backbone serving, and edge compatibility). Our technical contributions, of defining the gray-box setting and developing DGA/LGA under these strict access constraints, stand independently of any security claims.
> That said, the paper **already includes a discussion of security considerations**, specifically addressing gradient-based attacks. As noted in lines 191-198), we discuss that current gradient-inversion and extraction attacks are largely limited to simple architectures (e.g., single layer MLPs) and are *not known* to succeed on deep V&L backbones such as BLIP, CLIP, or DINOv2, as used in the paper. That said, the gray-box setting already provides significantly reduced internal visibility compared to LoRA or adapter-based methods, which reveal explicit weight updates and, as shown in prior work, enable weight reconstruction attacks. Because our methods do not modify/expose internal parameters, they inherently restrict the information available to an adversary.
>
> Having said that, we leave the deeper investigation of gradient-based attacks on sealed large backbones as future work, as we do not claim that our adapters are fully attack-proof. We will strengthen the discussion by clarifying the threat model and summarizing the constraints that already limit adversarial visibility in the gray-box setting.
> ## 2. Performance Under Significant Domain Shifts
> We fully agree. **We point and discuss this limitation in the main text** (Section 4.4 lines 432-443 and Section 6 in lines 483-485), including the reviewer’s comment on *“internal weight adjustments are still necessary for radical domain adaptation”* as a conclusion for this experiment.
> ## 3. Positioning Against Recent Black-box Prompt Optimization
> We thank the reviewer for their suggestion for Appendix D. Although it introduces this discussion, we agree that a more direct empirical comparison would make the argument stronger. However, due to computational limitations, and because 16-shot classification is only one instance within the much broader range of tasks and modalities we evaluate under the GrayBox setting, we chose not to run dedicated comparisons against these prompt-optimization attacks. We will clarify this in the revision and strengthen the discussion in Appendix D to better position Gray-box methods relative to black-box prompting approaches.
> ## 4. Computation and Training Efficiency
> We thank the reviewer for their suggestion. Our method does not claim any advantage in training efficiency over existing PEFT approaches, and we agree this is not a weakness nor advantage of the work. Our framework is motivated by access and deployment constraints, not by faster training. We will better clarify this explicitly in the revision.
>
>
> We thank Reviewer ka9w again for their thoughtful evaluation and for recognizing the breadth and rigor of our empirical study. We appreciate the constructive feedback and will incorporate the clarifications above into the final version to further strengthen the paper.

---

### Official Review · Reviewer_Rb73 · 2025-10-31

**Soundness:** 2
**Presentation:** 3
**Contribution:** 2
**Rating:** 2
**Confidence:** 4

**Summary:**

This paper introduces "Gray-box" fine-tuning, a lightweight, deployment-friendly framework for adapting foundation models without modifying internal parameters or architecture. It aims to address the challenges of deployment complexity, edge device optimization, and the exposure of proprietary model parameters. The method uses compact, external input/output adapters, trained with controlled gradient signals at predefined model entry points. Two variants are proposed: DarkGray-Box Adaptation (DGA), which only modifies input and output interfaces, and LightGray-Box Adaptation (LGA), which allows limited injection of learnable tokens at intermediate layers. Extensive evaluations show that Gray-box fine-tuning achieves competitive performance while maintaining strict constraints.

**Strengths:**

1. The paper is clearly written.

2. The paper introduces a novel setting, "Gray-box fine-tuning," which lies between black-box tuning and white-box tuning.

3. The experiments conducted are extensive and well-executed.

**Weaknesses:**

1. I don’t fully understand the necessity of Gray-box fine-tuning. Is it really needed in real-world scenarios?

2. If the setting of Gray-box fine-tuning holds, then the proposed method is essentially the simplest approach, which doesn’t seem particularly innovative.

3. Since the method allows for the introduction of additional learnable tokens in the intermediate layers, existing techniques like LoRA and Adapter also fit within the proposed LightGray-Box Adaptation framework. From this perspective, the LightGray-Box Adaptation (LGA) method proposed seems meaningless.

4. The paper mentions scalability, efficient edge deployment, and security, but how do DGA and LGA ensure these benefits? It appears that both introduce additional computational complexity, and LGA requires prior knowledge of the model, such as dimensions, for setting the fine-tuning locations. Doesn’t this impact the model’s privacy?

**Questions:**

N/A

---

> ### Author Response · Authors · 2025-11-27
>
> We thank the reviewer for the clear summary and for acknowledging the novelty of the Gray-box setting and the strength of the experimental evaluation. Below we address the concerns raised.
> ## 1. Necessity and Real-World Relevance of Gray-box Fine-tuning
> Gray-box fine-tuning is motivated by actual constraints emerging in commercial and regulated deployments, as we explicitly mention in lines 034-050. Specifically one interesting example: it is hard to maintain thousands of internal adapter modules or patched parameter states on a single backbone, which requires complex loading/offloading management. Some papers tackle this specifically for LoRA [1].
> ## 2. Simplest approach
> The simplicity is intentional. Our experiments show that even under the GrayBox constraints, meaningful adaptation is achievable. We would like to clarify that our key contribution (as we summarize in lines 130-145) are the GrayBox frameworks themselves, their analysis and demonstrations, rather than the adapter architectures which are the simplest we could think of. We agree that future work would optimize the adapter architecture. Interestingly, we note (in lines 258-262) that non-linearities do not necessarily improve results. We agree that improving the adapter architecture is a solid future work.
> 3. “LoRA/Adapter also fit within LGA; therefore LGA is meaningless.”
> Let us re-emphasize that LoRA and Adapter do not satisfy Gray-box assumptions:
> - They modify internal weights or add new internal layers (low-rank W updates, bottleneck MLPs).
> - They require access to all internal tensors and layer shapes.
> - They change the forward computation graph - input data should be processed at each LoRA layer before getting inserted to the next layer.
>
>
> LGA only injects tokens (as extra “visual patch” or “textual token”) into attention **without modifying any internal parameters**, and the tokens do not alter layers weights, but get processed by them as a regular input. This enables the backbone to remain sealed. We further justify this in lines 173-179.
> 4. Scalability, Edge Deployment, and Security Benefits
> Scalability: DGA/LGA support single-backbone multi-task serving: all task-specific components are external, eliminating the need to merge/swap internal adapters (a known bottleneck in serving LoRA at scale, Sheng et al., 2023, as we cite in lines 064-074 and 042-044). This drastically simplifies serving and improves batch-level efficiency. Potentially, the external adapters are simple and might be hosted on edge devices (e.g., phones) - serving as “pre/post-processing” stage to the backbone.
> Edge deployment: Edge compilers (e.g., ONNX) often prohibit dynamic internal structure modification. Since Gray-box methods leave the backbone untouched, one optimized backbone can serve all tasks, while input/output transformations remain inexpensive.
> Security / Privacy: Crucially:
> - DGA reveals only the gradient at the boundary, not internal activations or weights.
> - LGA exposes layer input shapes only, not their internal operations or parameters.
> - LoRA/Adapters, in contrast, leak enough information to reconstruct original weights [2].
> We further elaborate on these comparisons in lines 064-074 and Table 1.
> Computational cost: DGA adds <1% FLOPs on BLIP/CLIP, and LGA adds only O(k·d) token overhead. Measured FLOPs are included in Appendix D.
>
>
> We appreciate the reviewer’s comments and hope that this response clarifies their constructive points.
>
> [1] Ying Sheng, Shiyi Cao, Dacheng Li, Coleman Hooper, Nicholas Lee, Shuo Yang, Christopher Chou, Banghua Zhu, Lianmin Zheng, Kurt Keutzer, Joseph E. Gonzalez, and Ion Stoica. S-LoRA: Serving Thousands of Concurrent LoRA Adapters. CoRR, abs/2311.03285, 2023
>
> [2] Eliahu Horwitz, Jonathan Kahana, and Yedid Hoshen. Recovering the pre-fine-tuning weights of generative models. arXiv preprint arXiv:2402.10208, 2024.

---

### Official Review · Reviewer_Q4xG · 2025-10-31

**Soundness:** 2
**Presentation:** 2
**Contribution:** 2
**Rating:** 4
**Confidence:** 4

**Summary:**

This paper introduces Gray-box fine-tuning, a novel paradigm for adapting foundation models under restricted internal access, aiming to balance adaptability, deployment efficiency, and model security/IP protection.
Two variants are proposed:

DarkGray-Box Adaptation (DGA): trains lightweight input/output adapters while keeping all backbone weights and architecture frozen.

LightGray-Box Adaptation (LGA): further allows the injection of a few learnable tokens at intermediate layers without altering internal weights.

Extensive experiments across multiple modalities — including text-to-image retrieval, video retrieval, image classification, sketch retrieval, and diffusion-based generation — demonstrate that the proposed Gray-box methods achieve reasonably competitive performance compared with conventional fine-tuning approaches.

**Strengths:**

1. The authors propose a new adaptation concept, Gray-box adaptation, which aims to minimize structural and parametric modifications during fine-tuning. This design simplifies the adaptation pipeline and makes it particularly suitable for privacy-sensitive or proprietary model deployment scenarios, showing clear practical relevance.

2. The experimental section is comprehensive. The authors evaluate the approach on multiple backbones and diverse datasets, presenting convincing results that strengthen the paper’s empirical soundness.

3. The writing and organization are clear and logical, with well-defined sections and consistent argumentation throughout the paper.

**Weaknesses:**

1. While the proposed Gray-box adaptation indeed reduces model access and helps mitigate issues like model thievery, the claimed simplification over existing approaches (e.g., Prefix/Prompt Tuning, LoRA, or Last-Layer Fine-Tuning) is not convincingly demonstrated. The paper lacks concrete comparisons in FLOPs, latency, or deployment complexity to support its efficiency claims.

2. Methodologically, DGA mainly performs linear transformations on the input/output spaces, and LGA adds learnable tokens at intermediate layers. However, its mechanism closely resembles existing prompt or prefix-tuning approaches (e.g., MaPLe), making the degree of novelty somewhat questionable.

3. The ablation study is limited. It only analyzes the effects of visual and textual adapters, without exploring other critical factors (e.g., number or position of learnable tokens, adapter capacity, or gradient access scope).

4. In many experiments, DGA and LGA underperform compared with LoRA or full fine-tuning, suggesting that simple linear transformations at the input/output level might be insufficient for effective adaptation. It would be valuable to design more expressive or adaptive Gray-box modules to further enhance performance while maintaining restricted model access.

**Questions:**

N/A

---

> ### Author Response · Authors · 2025-11-27
>
> We thank reviewer Q4xG for the constructive feedback and for recognizing the practical relevance, clarity, and breadth of our work.
> ## Efficiency and Deployment Simplification Claims
> We conducted FLOPs comparison in appendix D. However, our efficiency claims center on deployment complexity rather than FLOPs alone. Gray-box methods avoid (i) modifying internal weights, (ii) inserting modules inside the backbone, and (iii) maintaining per-task internal adapter routing. This enables single-backbone serving: all task-specific components are placed externally, and are applied before/after the backbone forward pass, which remain intact in inference.
>
> 2. Novelty Compared to Prompt/Prefix Tuning/MaPLe
> We agree that LGA resembles prompt-style methods, but the **core novelty lies in the strict access constraints and the corresponding problem formulation**, not in our adapter architectures themselves. However, prior prompt/prefix methods perform on single modality NLP tasks using 10-200 learned tokens (while we use 1-2) and MaPLe insert per-layer MLPs or trainable prefixes inside every transformer block, which also requires per-layer access. We state these differences in lines 149-190.
>
> 3. Scope of Ablation Studies
> We respectfully disagree that our ablation study is limited. Our paper includes extensive ablations: we conduct ablation studies on the number of input tokens in DGA and the choice of layers in LGA in Appendix B, in addition to the detailed adapter ablations in Section 5. We appreciate the request for deeper analysis. Regarding “gradient access scope” - the GrayBox framework assumes gradient access to the model as we define in lines 57-61. Regarding “Adapter Capacity” - in our paper, we conduct experiments across multiple backbones and tasks (also in Appendix A) which includes LLM, Diffusion Model, MultiModal backbones CLIP/BLIP, and ViT DinoV2, which demonstrates the broad capacity of the graybox framework.
> 4. Performance Gap to LoRA / Full Fine-Tuning
> We emphasize that Gray-box methods are not designed to surpass white-box methods but rather to serve settings where such access is impossible due to IP or deployment constraints.
> Nonetheless:
> - In low-data or retrieval-focused settings (COCO subsets, Cars, MSR-VTT), DGA achieves results within 0.3–1.7 points of LoRA, and sometimes surpasses it.
> - The gaps appear primarily in cross-domain adaptation (Sketchy, ImageNet-Sketch), which we explicitly acknowledge as a limitation in our analysis. These tasks inherently require weight-level adaptation due to large distributional shifts.
>
>
> Designing More Expressive Gray-box Modules: We appreciate this forward-looking suggestion. Our goal in this submission is to establish the formulation, and we intentionally use the simplest possible modules (linear, tokens) to illustrate the viability of the restricted-access setting. We will add a paragraph on potential extensions (e.g., low-rank nonlinear adapters, cross-modal adapters) as future work.
>
> We believe these clarifications, additional quantitative comparisons, and expanded ablations will address the reviewer’s concerns and strengthen the paper.
>
> We thank the reviewer again for their constructive comments.

---

### Meta-Review · Area_Chair_ciaw · 2026-01-07

**Summary:**

This submission formalizes a “gray-box” adaptation regime for foundation models where the backbone is sealed (no weight edits and no internal architectural modifications), and task adaptation is performed via lightweight external input/output adapters (DGA) or by injecting a small number of learnable tokens at selected layers (LGA). Reviewers agreed the deployment and access-constraint framing is clear and practically motivated, and the empirical evaluation is broad across tasks, modalities, and backbones. However, multiple reviewers questioned the incremental nature of the proposed instantiations (especially LGA’s proximity to prompt/prefix-style methods), noted missing closest baselines and incomplete quantitative evidence for the claimed deployment simplicity/efficiency and security motivations, and highlighted clear limitations under large domain shifts where gray-box methods lag behind weight-updating PEFT (LoRA/full FT).

**Reviewer Concerns:**

The rebuttal helped on clarification and scope: the authors more clearly articulated that the main contribution is the access-constraint formulation (not claiming SOTA accuracy), pointed to FLOPs results and additional ablations (token count and layer choice), and acknowledged that large domain shifts remain a limitation where internal weight adaptation is often necessary.

That said, several core concerns remain outstanding: (i) novelty and positioning are still not fully convincing given the similarity of LGA to prompt/prefix-style adaptation, and the boundary between “gray-box” and existing paradigms feels partly definitional; (ii) the paper still lacks strong, quantitative deployment evidence (latency, serving complexity, concurrency, memory, routing/adapter management) beyond FLOPs and qualitative arguments; (iii) security/IP arguments remain largely narrative, with limited analysis of whether boundary gradients (DGA) or intermediate token injection (LGA) introduce extractability risks; and (iv) comparisons to the closest practical baselines remain contentious, and even if some methods violate the strictest constraints, the paper would benefit from a clearer empirical “tuning parity under comparable constraints” story and a concise cost-benefit summary across regimes.

**Reviewer Scores:**

Reviewer Q4xG: likely remains 4 (borderline), as the rebuttal clarifies scope/ablations but does not fully resolve novelty and deployment-quantification concerns.

Reviewer Rb73: may move from 2 to 3 (weak reject) given clarified motivation and constraint definition, but the “LGA overlaps with existing methods” and “benefits not demonstrated quantitatively” concerns largely persist.

Reviewer ka9w: likely remains 4, with main remaining requests being deeper security/threat-model discussion and clearer training-efficiency/positioning vs black-box prompt optimization.

Reviewer Rr9e: likely remains 4, with outstanding requests on closest baselines (or clearer constraint-based exclusion plus a stronger comparable baseline suite) and a more explicit trade-off table/analysis.

---

### Decision · Program_Chairs · 2026-01-26

Reject